



# The first rainfall erosivity database in Mexico: facing challenges of leveraging legacy climate data

Viviana Marcela Varón-Ramírez[1,2], Douglas A. Gómez-Latorre[2], Carlos Eduardo Arroyo-Cruz[1],
Alberto Gómez-Tagle[3], Blanca Lucía Prado Pano[4], Ronald R. Gutierrez Llantoy[5],
Deyanira Lobo-Luján[6], and Mario Guevara[1]

[1]Instituto de Geociencias - Universidad Nacional Autónoma de México - UNAM. Campus Juriquilla, Qro. Mexico.
[2]Corporación Colombiana de Investigación Agropecuaria - AGROSAVIA, Centro de Investigación Tibaitatá,
Mosquera-Cundinamarca, Colombia
[3]Instituto de Investigaciones sobre los Recursos Naturales - INIRENA, Universidad Michoacana de San Nicolás de Hidalgo.
Morelia, Mexico
[4]Instituto de Geología - Universidad Nacional Autónoma de México - UNAM. Mexico City. Mexico.
[5]Departamento de Ingeniería (GERDIS, GEOSED), Pontificia Universidad Católica del Perú. Lima, Peru
[6]Instituto de Edafología - Universidad Central de Venezuela. Maracay, Aragua. Venezuela

**Correspondence:** Viviana Marcela Varón-Ramírez (viviana.varon@geociencias.unam.mx) and Mario Guevara
(mguevara@geociencias.unam.mx)

**Abstract.** Soil Water Erosion (SWE) is the dominant soil degradation driver on a global scale. For quantifying SWE, erosivity is an index that reflects the potential (i.e., the energy) of rainfall to cause SWE. To enhance the assessment of the SWE process at the national scale—, the objectives of this research are a) to develop the first Mexican rainfall time series database for three climate normals CNs (1968-1997, 1978-2007, and 1988-2017) leveraging legacy climate data, and b) to estimate rain-
fall erosivity across continental Mexico by using daily rainfall time series. The workflow has three methodological moments: 1) development of the rainfall time series database, 2) estimation of rainfall erosivity, and 3) rainfall erosivity verification. First, we compiled and harmonized over 5000 useful rainfall time series (RTS) well distributed across the Mexican territory. We performed a quality assurance, homogeneity analysis (using the normal homogeneity test), and data gap-filling (using the proportion method). Then, we use a potential power law equation to estimate rainfall erosivity at daily resolution. Finally,
we compared and verified our results with three external datasets (global, national, and local scales). The principal research product is a new database with 1370, 1678, and 1676 RTS for each CN and its corresponding rainfall erosivity. The mean values for rainfall erosivity for the three CNs were 3600, 3296, and 3461 MJ mm ha$^{-1}$ h$^{-1}$ yr$^{-1}$, respectively. The statistical distribution of the erosivity values was right-skewed for the three CNs, with high erosivity values reaching >8000 MJ mm ha$^{-1}$ h$^{-1}$ yr$^{-1}$ in all the three CNs. About the verification of erosivity values, we found that Tropical rain-forests, temperate
Sierras, and the Great Plains are the ecoregions with more significant differences concerning the global database, a generalized underestimation of erosivity values concerning the national dataset, and an adjustment coefficient of 1.85 for a local condition in Michoacan state. This new database provides tools for daily climatological analysis across Mexican territory and through a multiyear period (1968 to 2017). Erosivity results trigger the study of SWE at the national scale by identifying areas with higher susceptibility to soil loss due to rainfall action and providing a more spatially dense erosivity database that





follows the pattern of erosivity databases from higher time resolution. Following the FAIR principles (Findability, Availability, Interoperability, and Reproducibility) for scientific data, this database is available from a scholarly accepted repository (https://doi.org/10.6073/pasta/7479676e406aeb40127da7b096b28eb2) for public consultation.

Keywords: legacy climate data, gap filling of climate series, daily rainfall erosivity, climatol package

## 1  Introduction

Soil Water Erosion (SWE) refers to the soil displacement from its original location due to water action, such as rainfall, overland flow, and irrigation (Nearing, 2013). SWE represents the dominant soil degradation issue at the global scale because it affects nearly 33% of the World's surface (Pennock, 2019). The impact of SWE is not just on-site but also has off-site effects on distant locations. On-site, soil loses its natural fertility and capacity to store water, nutrients, and organic carbon (Hatfield et al., 2017), affecting food security. Off-site, the eroded soil triggers environmental issues such as water pollution,
dam siltation, eutrophication of water bodies, contamination of coastal and marine ecosystems, and overall environmental damage (Feng et al., 2023).

When other erosion factors (e.g. erodibility, soil coverage and management, and topography) are constant, regions with frequent rainfall experience more soil loss than areas with limited rainfall (Ke and Zhang, 2021). Rainfall erosivity is the potential of rainfall to cause SWE (Nearing et al., 2017). Furthermore, rainfall erosivity is the first-factor influencing SWE and
is crucial for soil and water conservation planning. As extreme rainfall events are expected to increase in tropical zones due to climate change, SWE will likely increase as well (Borrelli et al., 2020). For example, in Mexico, the temporal distribution of rainfall has become more extreme, with more extended periods of drought and increasingly extreme rainfall events (Porrúa et al., 2020). Thus, understanding rainfall erosivity patterns across Mexico is essential for enhancing soil sustainability and informed decision-making in soil conservation

Rainfall erosivity—often represented as the $R$ factor— quantifies the potential of rainfall to cause SWE (Nearing et al., 2017). The $R$ factor captures the combined effect of raindrop impact and water flow. This $R$ factor, when calculated by rainfall event as the product of total storm energy ($E$) and the maximum 30-minute intensity ($I30$) is notated as $R(EI30)$ factor. However, $I30$ necessitates high temporal resolution rainfall data, often at a minute temporal resolution, which can be challenging to obtain, especially at a national scale of Mexico. However, when coarser time intervals are used instead of minute
temporal resolution, the relationship between $E$ and $I30$ remains consistent, but a more larger calibration coefficient is needed (Tu et al., 2023). For this reason, the finer the time interval of the rainfall time series (RTS) used to calculate the $R$ factor, the closer the estimations are to the $R(EI30)$ values as originally defined in Musgrave (1947).

In Mexico, the finest temporal resolution of legacy climate data to estimate rainfall erosivity is daily. As in many countries around the world, there are some issues in the available Mexican rainfall time series, such as missing values, short measurement
periods, and series inhomogeneity (breaks due to station relocation and measurement mistakes), which further compound the challenge of using climate data (McKinnon, 2022). In that way, erosion studies in Mexico have calculated the $R$ factor from coarse-resolution RTS (monthly or annual) for some specific regions (Benites et al., 2020; González et al., 2016). However,





a national estimation of the $R$ factor does not exist (Varón-Ramírez and Guevara, 2024). Thus, climatology studies, such as erosivity, need a complete and reliable rainfall time series database (Yozgatligil et al., 2013). Therefore, a whole scheme of quality assurance, gap-filling, and homogenization process of the rainfall time series is needed (WMO, 2023). Consequently, with a reliable rainfall database, it is possible to represent the actual rainfall characteristics in a particular region and allow soil erosion monitoring at the local and national scales in Mexico.

Developing a rainfall time series and, consequently, a rainfall erosivity database is challenging in both spatial and temporal terms. The large diversity of topographic conditions (i.e., two principal mountain ranges and a large latitudinal extent) and proximity with large water bodies from the Pacific Ocean and the Gulf of Mexico makes Mexico a contrasting scenario of rainfall patterns (Carrera et al., 2024) and its hydrological-related processes(e.g., rainfall erosivity). Accurate benchmarks for understanding typical climate conditions and characterizing climate trends require a rainfall database long enough to represent its corresponding climate normal (CN), i.e., a statistical product computed over 30 years of rainfall time series (World Meteorological Organization, 2017). The CNs are widely used to compare recent observations, create anomaly-based datasets, and provide context for future climate projections. Considering local patterns across different CNs, these characteristics will contribute to an unprecedented rainfall time series dataset to estimate rainfall erosivity in Mexico.

The lack of reliable and complete daily climate databases has not allowed the study of the impact of precipitation on the SWE process at a national scale in Mexico. Hence, the main objectives of this research are 1) to develop the first Mexican rainfall-series database for three climate normals (1968-1997, 1978-2007, and 1988-2017), leveraging on the availability of legacy climate data, 2) to estimate rainfall erosivity across continental Mexico by using daily rainfall time series. The next section describes the three methodological moments of this research: rainfall time series database development, rainfall erosivity estimation, and rainfall erosivity verification. Then, the results are presented accordingly. We present discussion section with a critical analysis of results against recent scientific literature. We finally present our conclusions section; and the availability of all the codes and resulting databases of this research. The new knowledge allows a better understanding and prediction of rainfall distribution and its associated processes in different regions of Mexico.

## 2 Methodology

The study area corresponds to the conterminous Mexico (1,948,170 km$^2$). The country is located between latitudes 14°W and 32°N and longitudes 86°W and 118°W. Because of its geographical location, the region exhibits complex topographic and climate features (de Anda Sánchez, 2020). Mexico has been clustered by seven first level ecoregions—which represent geographical units with characteristic flora, fauna, and ecosystems (Commission for Environmental Cooperation, 1997)—, namely: Mediterranean California, North American Deserts, Semi-arid Elevations, Great Plains, Tropical Rain Forest, Tropical Dry Forest, and Temperate Sierras. Each ecoregion occupies 1.3, 28.6, 11.8, 5.5, 14.2, 16.4, and 22.3 % of the total country area, respectively (Figure 1).

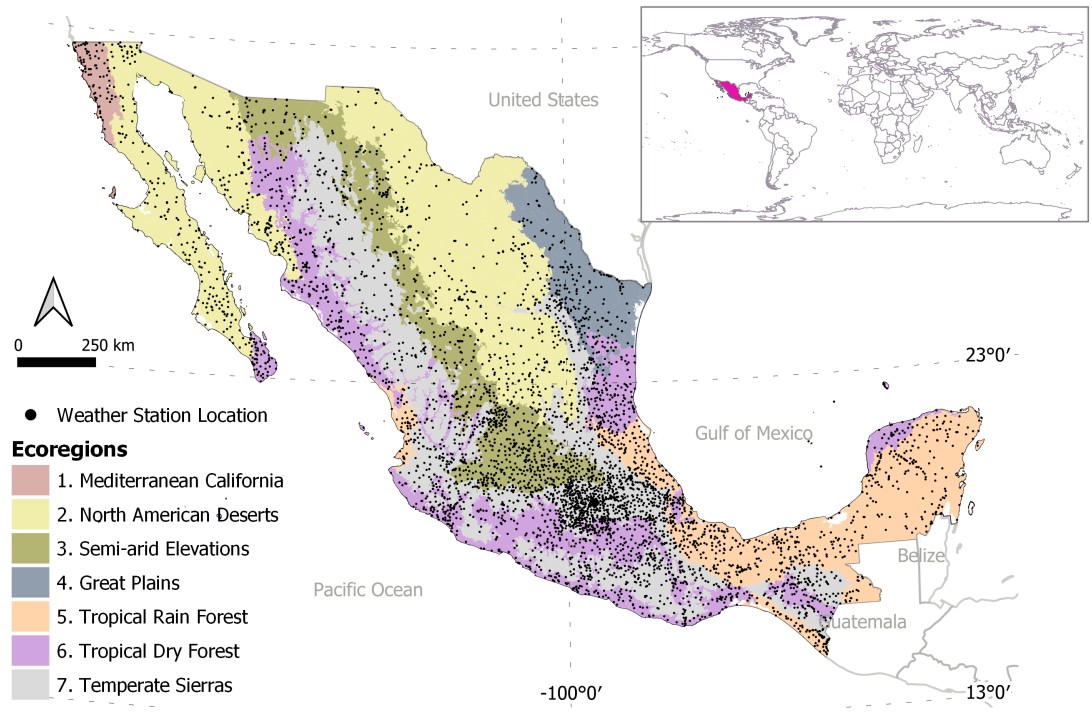

**Figure 1.** Ecoregions defined by the Commission for Environmental Cooperation (1997) and locations of the weather station available from the National Meteorological Service (SMN)

This research followed a workflow of three methodological stages (Figure 2). First, we developed a rainfall time series database with a daily resolution. Second, we estimated the rainfall erosivity by using daily rainfall time series. Third, we verified the rainfall erosivity estimations.

## 2.1 Rainfall time series (RTS) database development

The RTS database was developed through four steps: first, compilation, selection, and quality assurance of RTS. Second, the clustering of the RTS following its geographical and data attributes. Third, homogenization and data gap-filling of monthly and daily RTS. Fourth, quality control of the data gap-filling process.

### 2.1.1 Compilation, selection, quality assurance of RTS

In Mexico, the climate time series database results from the continuous effort of measuring, compiling, transcription, and analyzing data reported by weather stations distributed throughout the entire Mexican territory. This effort is led by the National Meteorological Service, which makes available to the public the raw data collected since 1900.





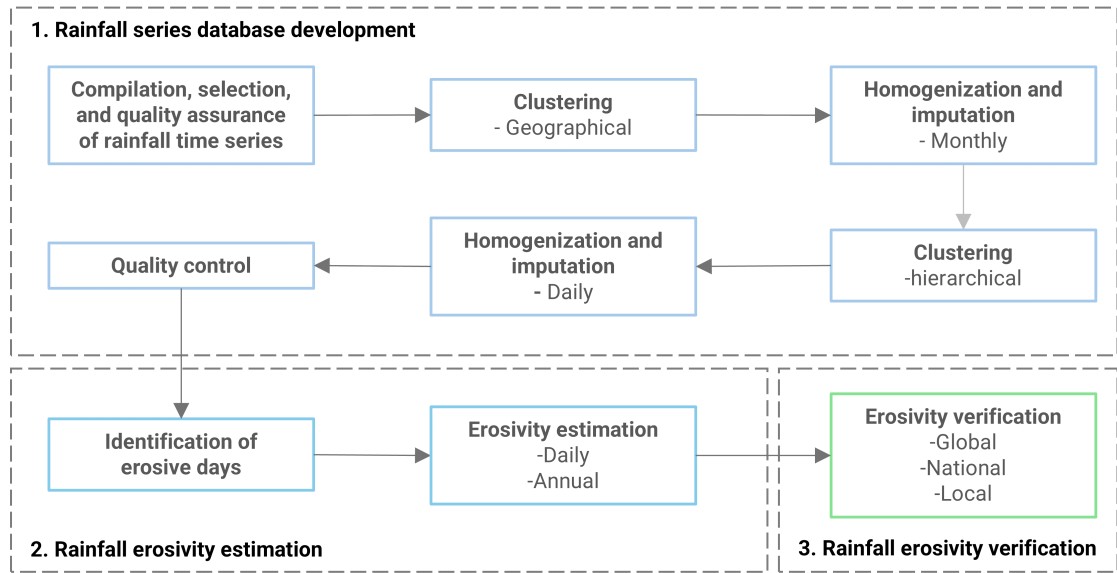

**Figure 2.** Workflow summarizing the three key methodological stages: (1) Development of the rainfall series database, (2) Estimation of rainfall erosivity, and (3) Verification of the calculated rainfall erosivity.

The data can be downloaded from the official National Meteorological Service site [https://smn.conagua.gob.mx/es/]. In the case of this project, 5454 files in plain-text format were downloaded, corresponding to the daily database of the entire network of weather stations (Fig. 1) available up to June 2022. Each file contains a header in which the weather station identification is reported and a series of data on its geographical location. Additionally, the files have daily data for rainfall, evaporation, and maximum and minimum temperature.

All data processing, including file downloading, was implemented in R (R Core Team, 2022). The download process was automated using the `download.file` function implemented in the `utils` package. A preparation pre-process facilitates the handling of data by separating the header's information and the daily data. The header data were extracted and converted to shapefile (sic) (.shp) format for spatial management. Subsequently, the file is converted in to a comma-delimited values format(*.cvs).

Finally, there were 5410 RTS at a daily temporal resolution with a unique format, and we considered those potentially useful RTS for our study. Consequently, there were two principal products in this step. First, a *.csv file with the information of the weather station, where each column corresponds to each of the data provided in the header (e.g., station ID, name, state, municipality, current situation, institution in charge, longitude, latitude, altitude, and report date), and each row corresponds to a weather station. Second, a set with 5410 *.csv files with the climate series of each weather station with four columns: date,

rainfall (mm), temperature (°C), and relative humidity (%).





To select the useful RTS, we identified the standard period of registers for each RTS between 1961 and 2018. On the other hand, for erosivity estimations, it is recommended to use a historical RTS of ≥20 years (Vantas et al., 2019; Renard and Freimund, 1994). Therefore, this article looked over the RTS in three climate normals: CN1 (1968 to 1997), CN2 (1978 to 2007), and CN3 (1988 to 2017). Subsequently, we selected those RTS with less than 20% of missing values (WMO, 2023) for each climate normal.

Calculating the monthly cumulative rainfall, we identified that long sequences of zeros and NAs were common. For quality assurance, we identified those years with no rainfall and replaced zero values with NAs. Those RTS with less than 20% missing values after the replacement were included in the resulting rainfall dataset. However, we used those RTS as a reference in the data gap-filling step.

### 2.1.2 Clustering of rainfall time series (RTS)

Clustering analysis is highly recommended when gap-filling a large set of RTS (Guijarro, 2014). Clustering similar RTS allows you to use information from related series to fill in gaps. Rainfall patterns often exhibit spatial and temporal correlations, so data gap-filling from a group of similar series can result in more accurate estimates (Fransiska et al., 2024). In this workflow, we performed the data gap-filling by clustering the RTS according to 1) the geographical space, and the environmental characteristics of Mexico and 2) the data dissimilarity among diverse environments (Figure 2). The first clustering was performed using the ecological regions of North America (Commission for Environmental Cooperation, 1997; INEGI-CONABIO-INE, 2008). The second separation was performed after the data gap-filling and homogenization of monthly series by ecoregion. A hierarchical cluster analysis groups the data series with similar seasonality and rainfall volume patterns (Gómez-Latorre et al., 2022). The number of clusters ($k$) ranged from 2 to $sqrt(n)$, where $n$ is the total number of stations in the dataset (Rohlf, 1974). The better $k$ values for each ecoregion were found using the Hartigan cluster validation index (Hartigan, 1975). This index was identified by Todeschini et al. (2024) to perform better when evaluating 68 cluster validation indexes (CVI´s) over 21 different datasets. However, we did not perform a clustering analysis for Mediterranean California and Great Plains ecoregions due to the small size of the ecoregion and, therefore, the amount of the RTSavailable for those ecoregions.

### 2.1.3 Homogenization and data gap-filling of RTS

We followed three steps to get a complete RTS for each CN: quality assurance, homogeneity analysis, and data gap-filling (WMO, 2020). In this step, quality assurance involved verifying the physical and statistical consistency of the series, discarding outliers whose standardized anomaly was outside a predefined threshold and was unrelated to any climate variability events. Outliers are removed and replaced as NA values to be completed in the data gap-filling processes.

Homogeneity analysis involves removing the biases caused by some artificial breaks in the RTS (Yan et al., 2014). These breaks result from common issues such as reading or instrumental mistakes, instrumental changes, or special situations at the weather station location Guijarro (2014). We used the standard normal homogeneity test (Alexandersson, 1986) to analyze homogeneity.



To fill in missing data, we used the proportions method. This method estimates the missing information based on neighboring stations, considering the distance between each station (Paulhus and Kohler, 1952). The procedure used the three precipitation

series with the highest correlation coefficient to the series that will be filled, with the condition of having been previously normalized. Then, we estimated $N_1 = \frac{A_1}{3}(\frac{N_a}{A_a} + \frac{N_b}{A_b} + \frac{N_c}{A_c})$; where $N_a$, $N_b$ and $N_c$ are the precipitation data for each of the stations with the highest correlation, while $A_a$, $A_b$ and $N_c$ are their corresponding normal average. All of the data gap-filling and homogenization process was made with `homogen` function of `climatol` package (Guijarro, 2024).

Appendix A1 presents a summary of the homogenization parameters used for each step. Table A1 shows the parameters

used to homogenize the monthly series, while Table A2 shows the parameters used to homogenize daily series by ecoregions 2, 3, 5, 6, and 7 and their corresponding subgroups.

### 2.1.4    Quality control of the data gap-filling processes

A quality validation was performed using the McCuen test (McCuen, 2016) to ensure consistency during the data gap-filling process of the rainfall time series. McCuen test compares the differences between the aggregated rainfall of the original multi-

annual monthly means and the final series. The generated RTS with a difference greater than 10% related to its original were discarded.

### 2.2    Rainfall erosivity estimation

The days with erosive rainfall were identified as those with cumulative precipitation greater than 12.5 mm as an extension of the suggestion by Wischmeier and Smith (1978); Shin et al. (2019); Efthimiou (2018). After, we calculated daily erosivity $R_d$

using the power law model proposed by Richardson et al. (1983); Alves et al. (2022); Beguería et al. (2018) with the sinusoidal relationship to describe the annual cycle of the coefficient of the power law function to represent seasonal differences in rainfall characteristics (Yu and Rosewell, 1996). Equation 1 presents the mathematical definition of $R_d$.

$$R_d = \alpha[1 + \eta cos(2\pi f j - \omega)]P_d^{\beta} \tag{1}$$

where $R_d$ is the daily erosivity, $P_d$ is the daily precipitation; $\alpha$, $\eta$, and $\beta$ are adjusted coefficients; $f$ is the monthly frequency

(1/12); $\omega$ is $7\pi/6$; and $j$ is the j-month of the year. The adjusted coefficients were adopted as in Xie et al. (2016), which utilized sixteen RTS at a one-minute temporal resolution to find the regression coefficients for the potential law equation.

The annual erosivity $R_y$ (MJ mm ha$^{-1}$ h$^{-1}$ yr$^{-1}$) was calculated as the sum of the daily erosivity values in a year time ($R_y=\sum_{i=1}^{ed} R_{d_i}$). The rainfall erosivity $R$ corresponds to the mean of the annual erosivity values for a multi-year period ($R=\sum_{j=1}^{30} R_{y_j}$). It is worth clarifying that, as we calculated the annual rainfall based on daily rainfall erosivity, estimated

with the equation of Xie et al. (2016), we denoted those values as $R(Xie et al. 2016)$. Finally, we obtained three datasets with rainfall erosivity values for the three climate normals: Mexico-CN1 (1968-1997), Mexico-CN2 (1978-2007), and Mexico-CN3 (1988-2017).





### 2.3 Rainfall erosivity verification

The rainfall erosivity values in the resulting datasets (Mexico-CN1, Mexico-CN2, and Mexico-CN3) were verified using three
databases at the global, national, and local scales. On a global scale, we used the GloREDa database (Panagos et al., 2023); at
the national scale, we used the database obtained by Torres (1991); and at the local scale, we calculated rainfall erosivity by
using 15-minute RTS in a mountain region in Michoacan state in Mexico.

On the global scale, the GloREDa database was built using data from almost 4000 weather stations worldwide (Panagos
et al., 2023), where 15 weather stations are widespread in the Mexican territory. They contain 5-minute RTS data from 2005 to
2015; however, not all the RTS have registers for the multiyear period of ten years. Indeed, some RTS have just four years of
registers. On the other hand, GloREDa provides monthly erosivity values in a raster format (.geotiff) (Figure 3a). We calculated
the rainfall erosivity of GloREDa database as the sum of the twelve raster files (one by month). We perrformed the verification
of our Mexico-CN3 and GloREDa dataset by comparing the mean and the standard deviation values by ecoregion.

At the national scale, Torres (1991) estimated rainfall erosivity using 54 RTS across Mexico (Figure 3b). The temporal
resolution for those 54 RTS is 1 minute, with a temporal period from 1977 to 1987. However, not all RTS have registers for
those ten years, so we selected 42 RTS presenting more than five years of registers. We will refer to this dataset as erosivity-
Cortés. Then, we adjusted a linear model using the R factor as the dependent variable and the multiyear mean rainfall as
the predictor variable. We adjusted those linear models with three datasets: erosivity-Cortés, Mexico-CN1, and Mexico-CN2
to compare the slope of the linear models. We mean-centered the predictor variable to shift its mean to zero, improving the
interpretability of the intercept.

At the local scale, the Michoacan mountain region has a database with 30 RTS (Figure 3c). These RTS have a 15-minute
temporal resolution, and the period of registers is from 2011 to 2017. The weather stations belong to the Association of
Producers, Packers, and Exporters of Avocado from Mexico (APEAM). For those RTS with more than five years of registers,
we estimated the rainfall erosivity $R$ as the mean value of the annual erosivity $R_y$ for the multiyear period. However, as finer
time resolution is available in this case, we estimated the rainfall erosivity $R(EI30)$ by rainfall event, as defined in Wischmeier
and Smith (1978) by using the `RainfallErosivityFactor` package (Cardoso et al., 2020) in R project. Afterward, we
aggregated the 15-minute RTS into daily totals by summing the values for each day. After we calculated rainfall erosivity as
described in section 2.2. As we calculated daily erosivity values using the adjusted parameters from Xie et al. (2016) we denoted
these rainfall erosivity values as $R(Xieetal., 2016)$. We identified the relationship between $R(EI30)$ and $R(Xieetal., 2016)$
by performing a linear regression. In this sense, the slope of the linear model quantifies the mean change of $R(EI30)$ when
$R(Xieetal., 2016)$ increases a unit.

### 3 Results

This section presents the results of the three principal methodological stages outlined in the workflow. First, we developed
a daily-resolution rainfall time series database, which provided a basis for further analysis. Second, we estimated rainfall

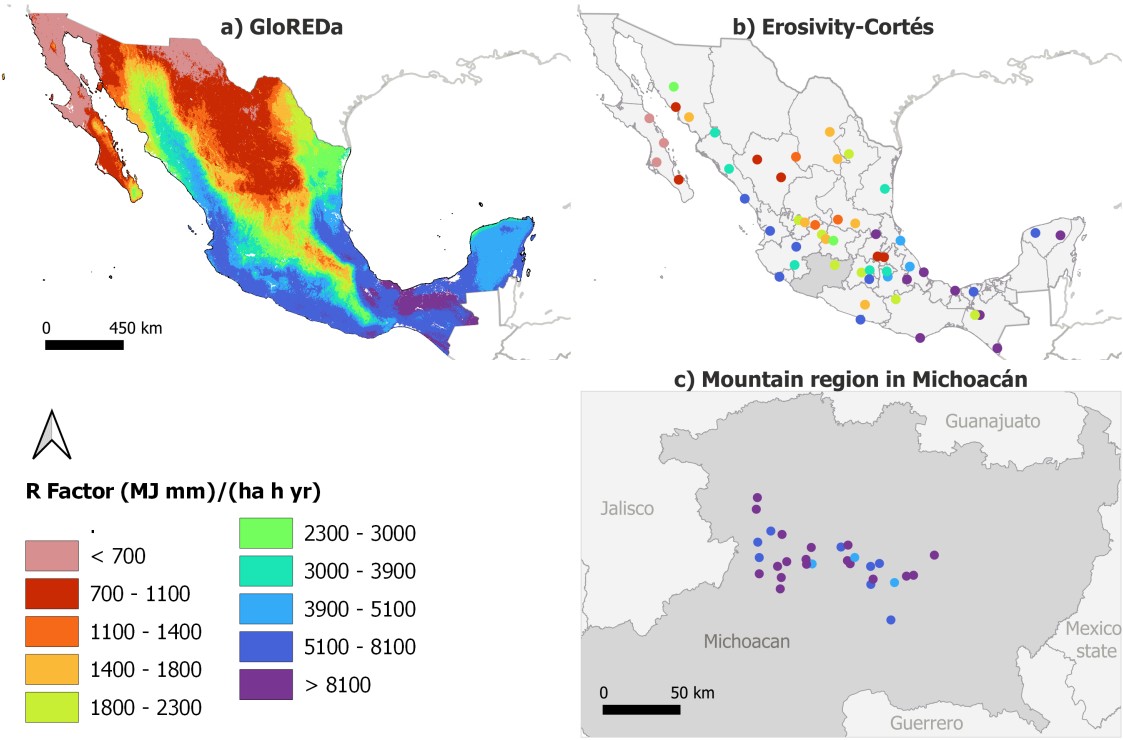

**Figure 3.** Spatial distribution of the validation datasets: a) GloREDa, b) Erosivity-Cortés, and c) Michoacan Mountain Region

erosivity values using a power law equation. Lastly, we verified these values by comparing them with global, national, and local databases.

### 3.1 Rainfall time series database development

The resulting database includes 1479, 1774, and 1721 RTS for CN1, CN2, and CN3, respectively (Table 1). The RTS with less than 20% of missing values and the identification of the number of series with NA and zero consecutive values are summarized

in Appendix A3. After the data gap-filling, homogenization, and quality control processes, we discarded the 4.8% of the RTS. Hence, Table 1 shows that the largest number of RTS corresponds to CN1 (1968-1997), with 7,4% (109 RTS), which is followed by CN2 (1978-2007), with 5,4% (95 RTS) and CN3 (1988-2017), with 1.9% (33 RTS). Likewise, the most significant proportion of discarded RTS corresponded to Mediterranean California (3 RTS, 21.4%) and North American Deserts (29 RTS, 15.84%) in CN1; Mediterranean California (2 RTS, 11.11%) and Great Plains (7 RTS, 12.72%) in CN2; and Great Plains (5

RTS, 10.86%) in CN3. Aditionally, in the data gap-filling processes, we identified that any of 5 RTS available for Mediterranean California had a rainfall register for three consecutive years; which did not allow us to carry out the data gap-filling process for the complete study period. Finally, the available number of RTS are 1370, 1676 and 1683 for CN1, CN2 and CN3, respectively. The available RTS are distributed across the Mexican territory for the three CNs and represent the seven ecoregions (Figure 4).



**Table 1.** Number of available rainfall time series before and after data gap-filling process, as well as the number of discarded rainfall time series. The frequency of rainfall time series (RTS) is shown by ecoregion and climate normal

| Ecoregion | RTS before data gap-filling process | | | RTS with change greater than 10% | | | RTS after the data gap-filling process | | |
|---|---|---|---|---|---|---|---|---|---|
| | 1968-1997 | 1978-2007 | 1988 - 2017 | 1968-1997 | 1978-2007 | 1988 - 2017 | 1968-1997 | 1978-2007 | 1988 - 2017 |
| Mediterranean California | 14 | 18 | 5 | 3 | 2 | | 11 | 16 | |
| North American Deserts | 183 | 273 | 243 | 9 | 18 | 4 | 154 | 255 | 239 |
| Semi-arid Elevations | 248 | 314 | 322 | 5 | 12 | 5 | 243 | 302 | 317 |
| Great Plains | 38 | 55 | 46 | 3 | 7 | 5 | 35 | 48 | 41 |
| Tropical Rain Forest | 195 | 237 | 236 | 6 | 6 | 3 | 179 | 231 | 233 |
| Dry Forest | 401 | 466 | 454 | 6 | 25 | 10 | 375 | 441 | 444 |
| Temperate Sierras | 400 | 411 | 415 | 27 | 25 | 6 | 373 | 386 | 409 |
| **Total** | **1479** | **1774** | **1721** | 109 | **95** | **33** | **1370** | **1679** | **1683** |

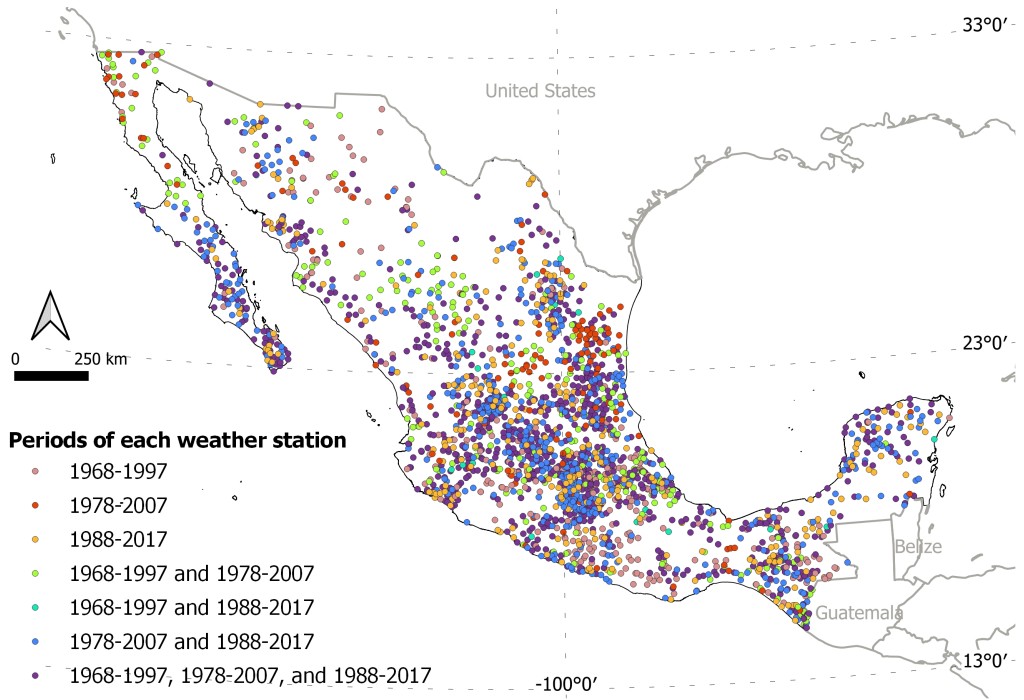

**Figure 4.** Spatial distribution of the weather stations

The percentage of variation of the series mean by ecoregion and CN using the two-step clustering is shown in Table 2. The
CN 1978-2007 showed the highest average change in the mean, with -1.80%, where it is also noted that four of the seven
ecoregions present relatively high averages changes. It is noteworthy that in CN 1968-1997, for North American Deserts, the
highest average change in the mean is -1.50%. In the period 1988-2017, for Tropical Dry Forest and for Temperate Sierras,





it is -3.38% and -1.61%, respectively. Additionally, in A4 there is shown the Root Mean Square Error (mm) of the monthly accumulated rainfall by ecoregion. It can be seen that the highest RMSE was found for Tropical Rain Forest in the three CNs

(6.12, 5.72, 6.78 mm, respectively), followed by Great Plains in CN2 and CN3 (5.72, 6.78 mm, respectively).

**Table 2.** Average percent (%) of change of the mean of rainfall time series by ecoregion

|  | Ecoregion | Average change of the mean (%) | | |
|---|---|---|---|---|
|  |  | 1968-1997 | 1978-2007 | 1988 - 2017 |
| 1 | Mediterranean California | -0.84 | 3.24 |  |
| 2 | North American Deserts | -1.50 | -1.36 | -0.14 |
| 3 | Semi-arid Elevations | -0.25 | -1.37 | 0.03 |
| 4 | Great Plains | -0.69 | 3.23 | -0.29 |
| 5 | Tropical Rain Forest | -0.06 | 0.20 | 0.10 |
| 6 | Tropical Dry Forest | -0.54 | -0.49 | -3.38 |
| 7 | Temperate Sierras | -0.20 | -3.73 | -1.61 |
|  | **Total** | -0.28 | **-1.80** | -0.98 |

Figure 5 shows the monthly rainfall distribution for the seven ecoregions for the multi-year period 1968-1997. In Appendix A1 is the monthly rainfall distribution for the period 1978-2007 and in the Apendix A2 is the monthly rainfall distribution for the period 1988-2017. Generally, rainfall is concentrated between May and November in ecoregions, except in Mediterranean California, where high values occur between November and April. For Mediterranean California, North American Deserts,

Semi-arid Elevations and Great Planins, the rainfall does not exceed 200 mm for the month with the highest volume. However, the rainfall in Tropical Rain Forest, Tropical Dry Forest and Temperate Sierras can reach up to 800 mm in June and September. These general behavior of rainfall distribution by ecoregion is the same for the three CNs.

## 3.2 Rainfall Erosivity estimations

In this section, we will display the results of the estimations of the annual erosivity values for the three CNs studied. First, a

descriptive statistic for erosivity values and their spatial distribution. Second, the identification of the erosive days.

The mean values for the three CNs expressed in MJ mm ha$^{-1}$ h$^{-1}$ yr$^{-1}$ were 3600 (SD 3748), 3296 (SD 3500), and 3461 (SD 3382), respectively (Table 3. The Krustal-Wallis test showed that the erosivity mean value for CN2 differed from CN1 and CN3 at a 95% confidence level. The coefficient of variation is around 1, which means significant variability. The statistical distribution of the erosivity values was right-skewed for the three CNs (skewness: 2.6, 2.9, 2.9, respectively), so the outliers

are high erosivity values reaching more than 8000 MJ mm ha$^{-1}$ h$^{-1}$ yr$^{-1}$ for the three CNs (Figure 6).

Regarding the spatial distribution, the erosivity values across Mexican territory look similar for the three CNs 7. The ranges plotted in the legend correspond (not precisely) to the deciles of the three statistical distributions. In the California peninsula, there are concentrated those fewer erosivities (peach dots) with less than 700 MJ mm ha$^{-1}$ h$^{-1}$ yr$^{-1}$, which represents the 1st decil of the distribution. The 2nd decil (values between 700 and 1100 MJ mm ha$^{-1}$ h$^{-1}$ yr$^{-1}$) is concentrated in the northern





**Figure 5.** Mean monthly rainfall for each ecoregion for the climate normal 1968-1997. a) Eco 1, b) Eco 2, c) Eco 3, d) Eco 4, e) Eco 5, f) Eco 6, and g) Eco 7.




**Table 3.** Descriptive statistic of the erosivity values for three climate normals (1968-1997, 1978-2007, 1988-2017). Min: minimum, Max:maximum, SD: standard deviation, CV: coefficient of variation, skew: skewness, kurt: kurtosis

| Climate normal | n | Mean | SD | Min | Max | CV | Skew | Kurt |
|---|---|---|---|---|---|---|---|---|
| **1968-1997** | 1369 | 3600a | 3748 | 48.0 | 31946.0 | 1.0 | 2.6 | 12.2 |
| **1978-2007** | 1678 | 3296b | 3500 | 39.0 | 29330.0 | 1.1 | 2.9 | 15.0 |
| **1988-2017** | 1676 | 3461a | 3382 | 47.0 | 28812.0 | 1.0 | 2.9 | 15.0 |

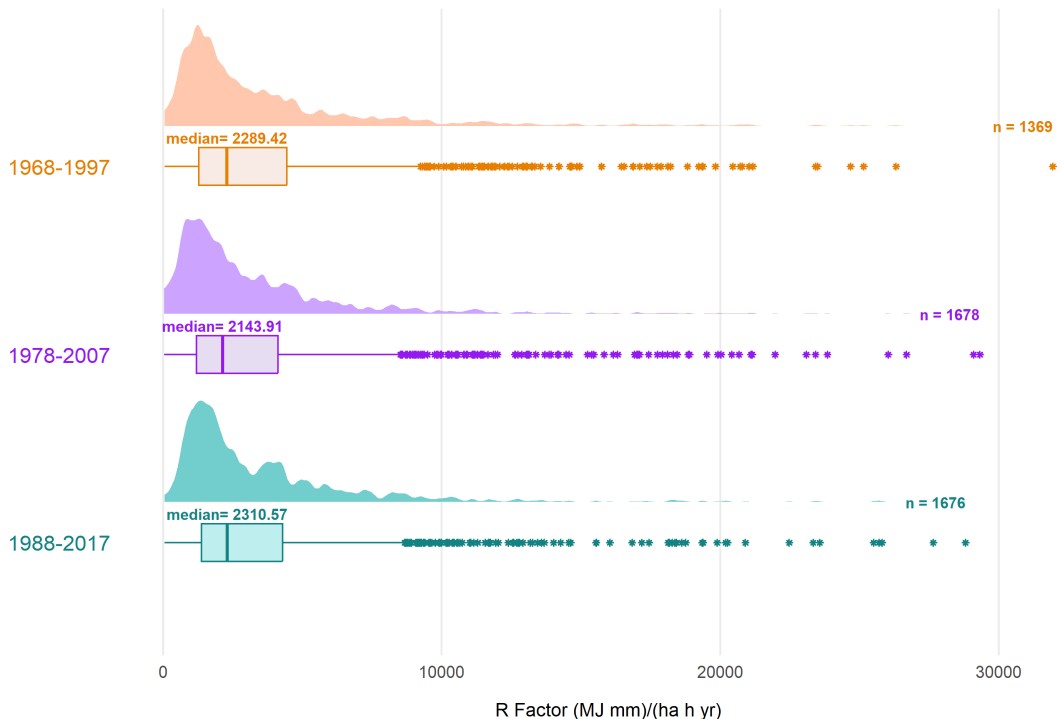

**Figure 6.** Density plot and Box-Plot of erosivity factor (R) calculated from daily rainfall time series for the three climate normals

and extends a little to the central region (red dots). These areas represent the North American deserts and Semiarid elevations ecoregions. On the other hand, the 9th (dark blue dots) and 10th (dark purple dots) with values between 5000 to 8100 and greater than 8100 MJ mm ha$^{-1}$ h$^{-1}$ yr$^{-1}$, respectively, are concentrated in the tropical rain forest ecoregion in the southwest region, and some places in the tropical dry forest ecoregion in the southeast of Mexico. It is clear that for the CN3 (1988-2017), there were fewer RTS for Mediterranean California and some places in the North American desert ecoregions because
the rainfall series there had records until 2012.

The erosive days were considered as those with rainfall greater than 12.5 mm. Figure A3 shows the mean number of locations with erosive rain for each day of the year. For the three CNs, there is one erosive period throughout the year; however, for CN3





the erosive period is marked by two peaks, first from day 173 to 190 and second from day 229 to 261; these days represent those with the most (10% higher) locations reporting erosive rainfall. On the other hand, the 90-decile number of locations

with erosive rainfall for CN2 and CN3 has increased by 7% (238) and 12% (227) concerning CN1 (212).

### 3.3 Verification of Rainfall Erosivity

In this section, we present the verification results of the usability of the national dataset of precipitation series at daily resolution. To achieve this, we compared the erosivity values of our datasets, called Mexico-CN3 and Mexico-CN2, with a global database (GloREDa), a national database (Erosivity-Cortés), and a local database (Erosivity-Michoacan) (Figure 8).

Using Mexico-CN3 and GloREDa datasets, we compared the mean values for six ecoregions (Figure 8a). Tropical rainforests, temperate Sierras, and the Great Plains are the ecoregions with more significant differences between means. Comparing Mexico-CN3 concerning GloREDa, there is an overestimation for Tropical Rain Forest and Great Plains, and for temperate Sierras, there is an underestimation. On the other hand, North American Deserts, Semi-arid Elevations, and Tropical Dry Forest ecoregions have very similar erosivity mean values. The reader must remember that CN3 (1988-2017) has no available

precipitation series in the Mediterranean California ecoregion.

At the national scale, comparing our datasets (Mexico-CN1 and Mexico-CN2) with the erosivity-Cortés, we identified that Mexico-CN1 and Mexico-CN2 well represent the relationship between $R$ and multi-year average annual rainfall (Figure 8b). When adjusting a linear model using $R$ as the dependent variable and the multi-year average annual rainfall as the predictor, we can find that the coefficients of the predictor variable are very similar, 5.95, 5.47, and 5.41, for erosivity-Cortés, Mexico-CN1

and Mexico-CN2, respectively. The intercepts (calculated with the predictor variable centered) are 3738, 3600, and 3296 for erosivity-Cortés, Mexico-CN1, and Mexico-CN2, respectively. The variance explained for each model is 0.64, 0.9, and 0.9 for erosivity-Cortés, Mexico-CN1, and Mexico-CN2, respectively.

At the local scale, the calculated $R(EI30)$ is greater concerning $R(Xieetal., 2016)$ as shown in Figure 8c. The coefficient for $R(Xieetal., 2016)$ is 1.82 (p-value 8.18e-10); The intercept is 8945 MJ mm ha$^{-1}$ h$^{-1}$ yr$^{-1}$ (p-value 2e-16); and the vari-

ance explained by the linear model is 84.7%. It is clear that the $R(Xieetal., 2016)$ values are underestimated compared with $R(EI30)$ when using Erosivity-Michoacan dataset.

### 4 Discussion

This research addressed the data incompleteness and breakpoints present in the available Mexican climate time series, resulting in a comprehensive and unprecedented database of rainfall time series. The new information is primarily applicable to the anal-

ysis of rainfall erosivity required for studying soil erosion at the national scale of Mexico. The new information is appealing for diverse users, as the insufficient availability of climate information (i.e., publicly funded data), it represents an interoperability barrier limiting the development of climate services and the understanding of key ecosystem climate-related processes at different scales (Vaughan et al., 2016). Consequently, scientists search for new ways to create national climate-related datasets containing state-of-the-art information for a diverse range of applications such as climate modeling, yield prediction, or eco-



logical forecasting. Particularly, we compiled and systematized a national dataset with daily rainfall and rainfall erosivity for three climate normals (1968-1997, 1978-2007, and 1988-2017) across Mexico. To the best of our knowledge, this research is the first effort to develop a large daily RTS dataset based on legacy climate data at the national scale.

We also address the data incompleteness and breakpoints that exist in Mexican climate time series available data, as indicated in previous reports (Cuervo-Robayo et al., 2020). The new information ensures the highest possible quality, (as explained

in the methods section) it includes a rigorous quality control, homogenization, and data gap-filling procedures of around 4000 daily RTS: 1370, 1679, and 1683 RTS for the CNs 1968-1997, 1978-2007, and 1988-2017, respectively. At the national scale, other extensive studies have been conducted, but they use a coarser time resolution compared to that studied here. For example, Cuervo-Robayo et al. (2014) updated the mean monthly rainfall for 100 years (1910-2009) using around 5000 RTS. Studies compare multiple gap-filling approaches at a regional scale (Cespedes et al., 2023) and conduct quality control and

homogenization process (García-Cueto et al., 2019) using daily RTS. Yet, those studies use RTS datasets representing shorter periods of time compared to our study. Studies using RTS across comparable periods of time to our study consider RTS with fewest missing data, without considering a data gap-filling process Pineda-Martínez and Carbajal (2017); Mateos et al. (2016). Thus, our research represents a significant contribution by offering a more comprehensive and higher-quality daily RTS dataset, addressing limitations related to the dataset size, representativeness, and the rigorous process of homogenization and

data gap-filling. We contribute with a new standard for climate data analysis at the national scale in Mexico.

Our methodology allowed us to reconstruct climate series with significant missing data gaps, as reported in previous studies (Guijarro, 2014). The methodology applied on an ecoregion basis, potentially increases spatial coherence in the completed and homogenized RTS, as previously recommended (Adeyeri et al., 2022). The methodology is also recommended by the World Meteorological Organization (World Meteorological Organization, 2020). Other robust data gap-filling strategies leverage high

computational capacity and big data analyses (i.e., machine learning) to obtain reliable RTS (Hırca and Türkkan, 2024; Lupi et al., 2023). These strategies build a predictive function to estimate values for missing data using the climate series data itself or auxiliary data such as satellite products (Duarte et al., 2022). Despite the performance predicting missing data values (Hırca and Türkkan, 2024; Lupi et al., 2023), many machine learning methods are still experimental, and a well-known and WMO-recommended methodology (as a reference point) is needed to benchmark more complex data-driven approaches. We recognize

that our effort is not error free, and therefore we present an estimation of error in our gap-filling approach in supplementary material A4. Future improvements of this new RTS database must include a progressive exploration of multiple gap-filling data techniques.

The gap-filled dataset (considering the completed and homogenized RTS for the three CNs) is useful for analysis of changes in precipitation trends (Yan et al., 2014) across Mexico. This new RTS dataset shows that the regional variation in rainfall

patterns is markedly latitudinal (Figure 5 and Appendix A1 and A2 ) , from Tropical Rain Forests across the south east, to the water limited environments of North American Deserts. These results are consistent with rainfall patterns previously reported by de Anda Sánchez (2020). In addition, the new dataset also provides an earlier perspective of the rainfall erosivity spatial distribution in Mexican territory.





Our work reveals that the distribution of erosivity values in Mexico corresponds to the geographical distribution of the rainfall
and seasonal rainfall conditions. The areas with the major erosivity values are concentrated in the isthmus of Tehuantepec (i.e.,
the shortest distance between the Gulf of Mexico and the Pacific Ocean), where the highest mean annual rainfall values in
Mexico have also been reported (de Anda Sánchez, 2020). In contrast, lower erosivity values are in Mexico's central and
northern regions, where severe droughts (e.g., that from the 1990s until the beginning of the twenty-first century), due to the
large-scale changes in ocean-atmospheric circulation patterns, have affected the mean annual rainfall. On the other hand, we
highlight a spot in the south of the California peninsula (e.g., Sierra La Laguna), where erosivity values are higher than those
estimated in the north. This local variation is due to the influence of tropical cyclones, which contribute up to 50% of the mean
annual rainfall in this region (Agustín Breña-Naranjo et al., 2015). Overall, the spatial variability in erosivity across Mexico
reflects the interplay between rainfall patterns and climatic events, underscoring the significant influence of regional weather
phenomena on soil erosion processes.

The new dataset is appealing for validating global datasets. Comparing the new dataset with GloREDa, we identified that
the mean values of rainfall erosivity for North American Desserts, Semi-arid Elevations, and Tropical Dry Forest ecoregions
are similar in more than 50% of the Mexican area. In contrast, those ecoregions with higher annual rainfall such as the Tropical
Rain Forest, show differences compared to GloREDa. This difference is evident because the ecoregion has higher mean annual
rainfall and standard deviation (862 to 4823 mm and an SD of 758 mm). In comparison, the GloREDa data set reports a lower
mean annual rainfall (1383-2100 mm and an SD of 402.35 mm, those values calculated with three RTS in the ecoregion). The
same pattern is observed for Temperate Mountains where Mexico-CN3 has a wider probability distribution of mean annual
rainfall (range of 318-4009 mm and an SD 500 mm) than GloREDa (range of 814 to 2258 mm, in this ecoregion GloREDa has
just two RTS). We therefore report a more complete description of RTS variance compared with that in GloREDa. We beleive
that our contribution could be useful to better represent global erosivity estimates.

Aligned with our results, Fenta et al. (2023) found the largest differences in annual rainfall erosivity values between the
GloREDa database and a satellite-based approach (using sub-hourly RTS) for the rainiest regions (tropical and temperate
climates) around the world. The best agreement between satellite-based rainfall erosivity (using the satellite precipitation
estimates corrected and reprocessed with the Climate Prediction Center Morphing Technique - CMORPH) and interpolated
GloREDa (Panagos et al., 2017) was found in Europe, where the density of rainfall gauges is the highest globally (Bezak
et al., 2022). Furthermore, with just 15 rainfall gauges in Mexico, it is important to identify the regions with high discrepancies
between national approach and interpolated GloREDa, in order to understand their limitations and use them in places with
scarce rainfall erosivity information.

At the national scale, we observe a general underestimation of erosivity values compared with the erosivity-Cortés dataset,
which is evident in the intercepts of the linear models (Figure 8b). Although calculated at a coarser temporal resolution, this
pattern has already been indicated by Tu et al. (2023); Yin et al. (2015) while evaluating the effect of modifying the time interval
for calculating $R(EI30)$ using 5, 15, 30, and 60-minute RTS. The authors concluded that increasing the time interval leads
to underestimating erosivity values. Similarly, Li et al. (2022) identifies that using a monthly model underestimates $R(EI30)$.
However, the same author found an overestimation of the R values regarding $R(EI30)$ using annual models. Therefore, having





more detailed information is arguably the best way to estimate the erosivity factor with greater certainty and to know which
model explains the greatest variance of $R(EI30)$. Our results highlight the advantage of having detailed information for the
mountain region in Michoacan.

Comparing our results with the erosivity-Michoacan dataset $R(EI30)$ based on a 15-minute RTS with the $R$ calculated
from Xie et al. (2016), we observe that for every unit change in $R$ with Xie´s equation, the $R(EI30)$ is 1.85 times larger. This
can be seen in the slope of the linear model in Figure 8c. The adjustment coefficient applies only to the rainfall conditions
of the mountain region in Michoacan. For other areas, it has been shown that the fit of a model is not the same for different
rainfall patterns Li et al. (2022). Furthermore, this adjustment may not be applicable across all Mexican territories. Additionally,
although Xie´s model has been tested for other hourly data sets and good results have been obtained (CC=0.96, NSE=0.91, and
BIAS=-1-11%), it is still recommended to use the data with the finest temporal resolutions, especially for those regions with
smaller amounts of rainfall (Chen et al., 2020). In Mexico the smaller amounts of rainfall fall in the ecoregions of California
Mediterranean, North American Deserts, and the Great Plains.

As potential limitations to this study, we selected just one methodology for gap-filling daily data. Multiple remote sensing
products could help to improve gap-filling efforts. These products include the Climate Hazards Group InfraRed Precipitation
with Station data (CHIRPS) (Funk et al., 2015) with daily temporal resolution; the National Oceanic and Atmospheric Ad-
ministration (NOAA) data presenting hourly time resolution, and Climate Prediction Center Morphing Technique (CMORPH),
among others. Using various data sources improves the robustness of the resulting datasets (Bessenbacher et al., 2023) because
those products enhance the completeness of the RTS and the accuracy of the gap-filling data. On another hand, the selection of
the model for calculating erosivity is a source of uncertainty in subsequent models estimating soil loss rates (Li et al., 2022).
Therefore, evaluating the performance of different adjusted models is appealing for future research when high-resolution time
series data are available. Additionally, at the national scale, it is important to test other daily models and estimate the variation
of the predictions by ecoregions. Additionally, if finer RTS were available for the entire national territory, it would be possible
to calibrate a potential equation for Mexico.

The new database is a potential tool for local, national and global erosion studies. At the local scale this database could serve
as a tool to design field experiments to validate rainfall erosivity estimations in different rainfall patterns, as well as to identify
the magnitude of the underestimations or overestimation when using different time-resolution of the RTS (Meng et al., 2021;
Zhao et al., 2019; Dunkerley, 2019). At the national scale, this database could serve as input in erosion models to generate a
base line of soil loss rates across Mexican territory. This necessity have been highlighted in Bolaños González et al. (2016).
The authors emphasize the necessity to estimate soil and organic carbon loss rates. Additionally, this could help to support
the development of environmental monitoring systems in Mexico. At the global scale, the results could serve as validation
benchmarks and increase the representativeness of global rainfall erosivity database such as GloREDa (Panagos et al., 2023)
or the Global Rainfall Erosivity database from Reanalysis and Satellite Estimates -GloRESatE- (Das et al., 2024). The new
information is appealing for the aforementioned efforts as Mexican territory does not have enough data representation due to
the lack of primary information. The new dataset and their possible improvements would allow for a more accurate estimation

of erosivity and, thus, the annual amount of soil lost due to rainfall. This implies a better understanding of soil resources, better territorial planning, better agricultural and land use management, and better land conservation programs.

## 5 Conclusions

We present an unprecedent rainfall erosivity database across Mexico. The research used legacy climate data to achieve erosivity across Mexican territory. However, the rainfall time series (RTS) contained a relatively large number of missing values. In order to increase RTS completeness, a gap-filling procedure was performed to obtain gap-free estimates for three climate normals. The new database let provide a more detailed insight of rainfall erosivity respect to global models. This study reveals that
the North American deserts and Mediterranean California are regions where the rainfall has less erosive power, while tropical rainfall forests have the highest rainfall erosivity. The new database is available for public consultation. This database is for researchers and students, technical assistants, decision-makers, and others users interested in rainfall erosivity patterns and trends. Additionally, all environmental studies in Mexico, where the rainfall process is needed at the daily resolution, may benefit from this dataset.

## 6 Code availability

Rproject scripts to reproduce the workflow described in this research is available at: https://zenodo.org/records/13830947 (Varón-Ramírez, 2024)

## 7 Data availability

Following the FAIR principles for scientific data, we published our resulting databases (Mexico-CN1 1968-1997, Mexico-
CN2 1978-2007, and Mexico-CN3 1988-2017) and the completed daily rainfall time series for the three climate normals in the Environmental Data Initiative (EDI) at https://doi.org/10.6073/pasta/7479676e406aeb40127da7b096b28eb2 (Varón-Ramírez et al., 2024) .

Rainfall erosivity databases contain eleven columns with the weather station location (code, coordinates, altitude, name, and ecoregion), the root means squared error (RMSE) of the data gap-filling process, rainfall erosivity, the accumulated number of
days with erosive rainfall, and the multiyear mean rainfall.

Daily rainfall time series databases contain 1369, 1678, and 1676 columns for the climate normal 1968-1997, 1978-2007, and 1988-2017, respectively. Each column corresponds to one rainfall time series.

**Figure 7.** R factor values calculated with the power law equation proposed by (Xie et al., 2016) at daily resolution for three climate normals





**Figure 8.** Verification of rainfal erosivity datasets (Mexico-CN1, Mexico-CN2, and Mexico-CN3) with three erosivity databases at different scales (global, national and local). a) Global verification: Comparison of mean values of rainfall erosivity from GloREDa and Mexico-CN3 datasets in six ecoregions of Mexico. b) National verification: Comparison of rainfall erosivity values from Erosivity-Cortés, Mexico-CN1, and Mexico-CN2 by performing a linear regression model using annual rainfall (mm) as predictor. c) Local verification: Linear regression model using rainfall erosivity values, R(EI30) and R(Xie et al.2016), using RTS (from Erosivity-Michoacan dataset) at two temporal resolutions, 15-minute and daily resolution, respectively.



# Appendix A

## A1 Homogenization parameters for monthly rainfall time series

**Table A1.** Homogenization parameters for monthly series. Eco: Ecoregion (1: Mediterranean California, 2: North American Deserts, 3: Semi-arid Elevations, 4: Great Plains, 5: Tropical Rain Forest, 6: Tropical Dry Forest, 7: Temperate Sierras); dz.max and dz.min (upper and lower): standard deviations to consider suspicious and anomalous data

| Climate normal | Ecoregion | inht | dz.max lower | dz.max upper | dz.min lower | dz.min upper |
|---|---|---|---|---|---|---|
| 1968 - 1997 | 1 | 15 | 8 | 10 | -8 | -10 |
| | 2 | 25 | 12 | 13 | -7 | -8 |
| | 3 | 40 | 10 | 11 | -8 | -9 |
| | 4 | 20 | 6 | 7 | -5 | -6 |
| | 5 | 30 | 8 | 9 | -6 | -7 |
| | 6 | 30 | 14 | 16 | -10 | -10 |
| | 7 | 35 | 12 | 13 | -7 | -8 |
| 1968 - 1997 | 1 | 15 | 10 | 10 | -10 | -10 |
| | 2 | 35 | 12 | 13 | -8 | -9 |
| | 3 | 35 | 11 | 12 | -8 | -9 |
| | 4 | 20 | 8 | 9 | -7 | -8 |
| | 5 | 50 | 8 | 9 | -7 | -8 |
| | 6 | 30 | 13 | 14 | -10 | -11 |
| | 7 | 40 | 9 | 10 | -8 | -8 |
| 1968 - 1997 | 1 | – | – | – | – | – |
| | 2 | 50 | 14 | 14 | -10 | -10 |
| | 3 | 60 | 14 | 14 | -8 | -8 |
| | 4 | 15 | 7 | 7 | -6 | -6 |
| | 5 | 55 | 8 | 8 | -7 | -7 |
| | 6 | 60 | 14 | 14 | -10 | -10 |
| | 7 | 100 | 12 | 12 | -8 | -8 |

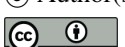



**Table A2.** Homogenization parameters for daily rainfall time series by ecoregion and group. Eco: Ecoregion (2: North American Deserts, 3: Semi-arid Elevations, 5: Tropical Rain Forest, 6: Tropical Dry Forest, 7: Temperate Sierras); RS: Number of rainfall time series; inht: ; dz.max and dz.min (upper and lower): standard deviations to consider suspicious and anomalous data

| | | | | 1968-1997 | | | | | | | | 1978-2007 | | | | | | | | 1988-2017 | | | |
|---|---|---|---|---|---|---|---|---|---|---|---|---|---|---|---|---|---|---|---|---|---|---|---|
| Eco | Group | RS | inht | dz.max lower | dz.max upper | dz.min lower | dz.min upper | Eco | Group | RS | inht | dz.max lower | dz.max upper | dz.min lower | dz.min upper | Eco | Group | RS | inht | dz.max lower | dz.max upper | dz.min lower | dz.min upper |
| 2 | 1 | 94 | 50 | 40 | 45 | -20 | -20 | 2 | 1 | 142 | 60 | 32 | 34 | -18 | -18 | 2 | 1 | 112 | 100 | 40 | 45 | -20 | -25 |
| | 2 | 18 | 25 | 40 | 45 | -25 | -30 | | 2 | 23 | 20 | 45 | 50 | -25 | -25 | | 2 | 12 | 10 | 35 | 40 | -25 | -30 |
| | 3 | 36 | 20 | 40 | 45 | -25 | -30 | | 3 | 68 | 12 | 35 | 40 | -30 | -35 | | 3 | 45 | 20 | 35 | 40 | -20 | -25 |
| | 4 | 35 | 20 | 35 | 40 | -20 | -25 | | 4 | 40 | 12 | 30 | 35 | -20 | -25 | | 4 | 67 | 50 | 45 | 40 | -30 | -35 |
| 3 | 1 | 34 | 25 | 24 | 26 | -10 | -12 | 3 | 1 | 41 | 30 | 22 | 24 | -12 | -14 | 3 | 1 | 115 | 60 | 26 | 28 | -16 | -16 |
| | 2 | 87 | 40 | 26 | 28 | -14 | -16 | | 2 | 55 | 25 | 22 | 24 | -14 | -14 | | 2 | 27 | 20 | 22 | 24 | -14 | -14 |
| | 3 | 59 | 40 | 22 | 24 | -12 | -12 | | 3 | 130 | 50 | 24 | 26 | -14 | -14 | | 3 | 180 | 40 | 26 | 28 | -18 | -18 |
| | 4 | 52 | 40 | 26 | 28 | -14 | -14 | | 4 | 71 | 40 | 24 | 26 | -12 | -14 | 5 | 1 | 80 | 60 | 22 | 24 | -20 | -22 |
| | 5 | 16 | 40 | 26 | 28 | -10 | -12 | | 5 | 16 | 30 | 22 | 24 | -10 | -12 | | 2 | 81 | 40 | 24 | 26 | -16 | -18 |
| 5 | 1 | 36 | 20 | 30 | 35 | -15 | -20 | 5 | 1 | 42 | 40 | 30 | 35 | -25 | -30 | | 3 | 75 | 100 | 24 | 26 | -16 | -18 |
| | 2 | 11 | 5 | 20 | 22 | -10 | -10 | | 2 | 13 | 50 | 18 | 20 | -10 | -12 | 6 | 1 | 290 | 200 | 35 | 40 | -25 | -30 |
| | 3 | 36 | 50 | 24 | 26 | -14 | -14 | | 3 | 30 | 40 | 18 | 20 | -12 | -14 | | 2 | 46 | 20 | 30 | 35 | -20 | -25 |
| | 4 | 58 | 50 | 30 | 32 | -14 | -16 | | 4 | 32 | 40 | 18 | 20 | -16 | -18 | | 3 | 85 | 80 | 35 | 40 | -30 | -35 |
| | 5 | 27 | 40 | 20 | 22 | -10 | -12 | | 5 | 68 | 50 | 22 | 24 | -16 | -18 | | 4 | 33 | 15 | 40 | 45 | -35 | -40 |
| | 6 | 27 | 40 | 20 | 22 | -12 | -14 | | 6 | 52 | 30 | 20 | 22 | -12 | -14 | 7 | 1 | 242 | 250 | 26 | 28 | -14 | -14 |
| 6 | 1 | 118 | 25 | 35 | 40 | -30 | -30 | 6 | 1 | 62 | 20 | 30 | 30 | -20 | -25 | | 2 | 59 | 100 | 35 | 40 | -25 | -25 |
| | 2 | 187 | 60 | 40 | 45 | -25 | -30 | | 2 | 267 | 80 | 30 | 35 | -25 | -30 | | 3 | 114 | 200 | 30 | 35 | -25 | -30 |
| | 3 | 77 | 25 | 40 | 40 | -20 | -25 | | 3 | 117 | 25 | 30 | 30 | -25 | -30 | | | | | | | | |
| | 4 | 19 | 25 | 50 | 55 | -30 | -35 | | 4 | 20 | 10 | 40 | 45 | -35 | -35 | | | | | | | | |
| 7 | 1 | 82 | 50 | 22 | 24 | -14 | -16 | 7 | 1 | 87 | 60 | 20 | 22 | -12 | -14 | | | | | | | | |
| | 2 | 49 | 50 | 35 | 35 | -20 | -25 | | 2 | 62 | 60 | 35 | 40 | -25 | -30 | | | | | | | | |
| | 3 | 153 | 50 | 35 | 35 | -25 | -25 | | 3 | 157 | 80 | 26 | 28 | -12 | -14 | | | | | | | | |
| | 4 | 116 | 50 | 24 | 26 | -10 | -12 | | 4 | 105 | 70 | 26 | 28 | -22 | -24 | | | | | | | | |





## A3 NA and zero values sequences identification

**Table A3.** Identification of number of rainfall time series with consecutive zero and NA values

| Consecutive years | Number of RS with sequences of NA and zeros | | | Decision |
| :---: | :---: | :---: | :---: | :---: |
| | **1968 - 1997** | **1978 - 2007** | **1988 - 2017** | |
| 1 | 794 | 787 | 611 | Not changed |
| 2 | 344 | 478 | 613 | Not changed |
| 3 | 155 | 286 | 277 | Replaced with NA and used as reference |
| 4 | 88 | 136 | 109 | Replaced with NA and used as reference |
| 5 | 77 | 78 | 82 | Replaced with NA and used as reference |
| 6 | 21 | 11 | 31 | Replaced with NA and used as reference |
| 7 | 1 | 1 | 0 | Removed |
| 8 | 2 | 0 | 1 | Removed |
| 9 | 1 | 2 | 2 | Removed |
| 10 | 2 | 0 | 0 | Removed |
| 11 | 3 | 2 | 1 | Removed |
| 12 | 0 | 0 | 0 | Removed |
| 13 | 0 | 1 | 1 | Removed |
| 14 | 0 | 1 | 0 | Removed |
| 15 | 0 | 0 | 0 | Removed |
| 16 | 1 | 2 | 0 | Removed |
| **RS (NA menor 20%)** | 1489 | 1785 | 1728 | |
| **RS removed** | 10 | 9 | 5 | |
| **RS for data gap-filling process** | 1479 | 1776 | 1723 | |





## A4 Root Mean Square Error (mm) of the data gap-filling process for the monthtly rainfall in the seven ecoregions

**Table A4.** Root Mean Square Error (mm) of the data gap-filling process by month and ecoregion. Eco: Ecoregion (1: Mediterranean California, 2: North American Deserts, 3: Semi-arid Elevations, 4: Great Plains, 5: Tropical Rain Forest, 6: Tropical Dry Forest, 7: Temperate Sierras)

| Climate Normal | Eco | Jan | Feb | Mar | Apr | May | Jun | Jul | Aug | Sept | Oct | Nov | Dec | Total |
|---|---|---|---|---|---|---|---|---|---|---|---|---|---|---|
| | 1 | 6.84 | 6.71 | 6.10 | 1.36 | 0.33 | 0.16 | 0.67 | 0.58 | 0.60 | 0.78 | 1.74 | 3.43 | 2.44 |
| | 2 | 1.91 | 1.39 | 1.02 | 1.31 | 1.37 | 2.20 | 6.32 | 3.42 | 3.55 | 2.84 | 1.11 | 2.41 | 2.40 |
| | 3 | 1.83 | 0.96 | 0.93 | 0.78 | 1.42 | 3.08 | 3.64 | 3.15 | 3.06 | 1.85 | 0.74 | 0.83 | 1.86 |
| **1968-1997** | 4 | 1.72 | 2.08 | 1.06 | 2.25 | 2.66 | 3.61 | 1.74 | 4.40 | 3.75 | 2.62 | 1.02 | 1.63 | 2.38 |
| | 5 | 3.07 | 3.57 | 4.28 | 3.20 | 6.06 | 7.92 | 8.39 | 8.45 | 9.79 | 8.32 | 5.66 | 4.78 | **6.12** |
| | 6 | 2.47 | 1.17 | 1.49 | 1.46 | 2.75 | 6.26 | 6.76 | 5.84 | 7.13 | 3.90 | 2.34 | 1.89 | **3.62** |
| | 7 | 2.31 | 1.07 | 1.53 | 2.06 | 2.75 | 4.55 | 5.11 | 5.00 | 8.03 | 3.93 | 2.11 | 2.06 | 3.38 |
| | 1 | 8.16 | 5.81 | 7.87 | 1.63 | 0.86 | 2.53 | 0.41 | 1.40 | 0.70 | 1.73 | 3.27 | 5.69 | 3.34 |
| | 2 | 2.13 | 1.13 | 0.86 | 1.38 | 1.73 | 1.98 | 5.26 | 3.95 | 3.84 | 2.10 | 1.06 | 2.28 | 2.31 |
| | 3 | 1.66 | 0.93 | 0.75 | 0.72 | 1.38 | 3.54 | 3.73 | 3.21 | 3.49 | 1.81 | 0.75 | 0.95 | 1.91 |
| **1978-2007** | 4 | 2.77 | 1.95 | 1.97 | 3.44 | 3.27 | 5.06 | 6.15 | 5.68 | 9.01 | 5.08 | 1.55 | 3.40 | **4.11** |
| | 5 | 4.04 | 3.82 | 3.15 | 3.02 | 6.01 | 7.71 | 6.34 | 7.07 | 8.66 | 9.14 | 5.22 | 4.51 | **5.72** |
| | 6 | 3.19 | 1.12 | 0.92 | 1.68 | 2.89 | 6.83 | 5.71 | 5.71 | 8.58 | 4.85 | 2.38 | 2.38 | 3.85 |
| | 7 | 2.37 | 1.31 | 1.17 | 2.35 | 4.08 | 5.61 | 6.34 | 5.54 | 6.92 | 6.92 | 2.72 | 2.27 | 3.97 |
| | 2 | 1.59 | 1.45 | 2.08 | 1.11 | 2.02 | 2.14 | 6.21 | 3.96 | 4.88 | 2.53 | 1.89 | 2.69 | 2.71 |
| | 3 | 1.67 | 1.68 | 1.23 | 0.77 | 1.41 | 3.23 | 4.29 | 4.56 | 3.21 | 1.99 | 1.05 | 1.11 | 2.18 |
| **1988-2017** | 4 | 2.75 | 2.53 | 4.08 | 6.08 | 8.74 | 5.68 | 13.17 | 9.87 | 12.52 | 6.27 | 5.05 | 6.27 | **6.92** |
| | 5 | 5.25 | 3.48 | 3.21 | 3.42 | 6.81 | 7.79 | 8.51 | 9.69 | 9.06 | 12.11 | 7.34 | 4.73 | **6.78** |
| | 6 | 3.56 | 1.68 | 1.57 | 2.38 | 3.48 | 6.35 | 6.55 | 6.78 | 8.28 | 5.42 | 2.41 | 2.36 | 4.24 |
| | 7 | 2.73 | 2.28 | 2.55 | 2.21 | 6.34 | 6.79 | 8.65 | 7.59 | 8.34 | 6.08 | 4.37 | 3.20 | 5.09 |
| **Total** | | 3.10 | 2.31 | 2.39 | 2.13 | 3.32 | 4.65 | **5.70** | **5.29** | **6.17** | 4.51 | 2.69 | 2.94 | **3.77** |





## A5    Monthly rainfall distribution for two climate normals (CNs) 1978-2007 and 1988-2017

**Figure A1.** Mean monthly rainfall for each ecoregion for the climate normal of 1978-2007. a) Mediterranean California, b) North American Deserts, c) Semi-arid Elevations, d) Great Plains, e) Tropical Rain Forest, f) Tropical Dry Forest, g) Temperate Sierras



**Figure A2.** Mean monthly rainfall for each ecoregion for the climate normal of 1988-2017. a) North American Deserts, b) Semi-arid Elevations, b) Great Plains, d) Tropical Rain Forest, e) Tropical Dry Forest, f) Temperate Sierras





## A6 Annual distribution of the number of locations with erosive rainfall



(a)

(b)

(c)

**Figure A3.** Number of locations with erosive rainfall of each day of the year for the three climate normal, a) 1968-1997, b) 1978-2007, and c) 1988-2017





*Author contributions.* Varón-Ramírez, Gómez-Latorre, and Guevara contributed with conceptualization, formal analysis, Methodology, and visualization. Varón-Ramírez, Gómez-Latorre, and Arroyo-Cruz contributed to the data curation and writing - original draft preparation. Arroyo-Cruz, Gómez-Tagle, Prado, Lobo-Lujan, Gutierrez, and Guevara contributed to Project administration and writing - review and editing. Prado and Guevara contributed to funding acquisition. .

*Competing interests.* The authors declare that they have no conflict of interest.

*Acknowledgements.* The authors want to thank to the National Meteorological Service (SMN by its initials in Spanish) and National Water Commission (CONAGUA by its initials in Spanish) for making public available the national climate data. Viviana Varón-Ramírez, Blanca Prado, and Mario Guevara acknowledges support from grant CF-2023-I-1846 of the National Council of Humanities, Sciences, and Technologies; abbreviated CONAHCYT. Mario Guevara acknowledges support from grant IGCP-765-UNESCO.



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
