# Peer review of "The first rainfall erosivity database in Mexico: facing challenges of leveraging legacy climate data"

_Earth System Science Data, 2024_

## Author Comment (AC1)

We appreciate the recognition of the value of this new database. We have addressed the necessity for rainfall information with a previous quality control. Since Mexico does not have a high-temporal rainfall monitoring system, at least not publicly available, we have compiled the most detailed rainfall information (daily resolution) and provided a valuable application of this new database: rainfall erosivity. We are grateful for the reviewers' comments, which have given us insights into how to present our research better. Following, we explain how we are going to address each reviewer's suggestion:

**Referee comments 1**

**The rainfall erosivity factor (R) is a crucial parameter for predicting water erosion, and its reliable estimation is essential for accurate water erosion assessments. However, the calculation of R is challenging due to the need for high-resolution rainfall process data (e.g., one-minute, five-minute, or ten-minute intervals), which makes it difficult to estimate accurately. The study uses daily precipitation data from 5,410 sites in Mexico, which is a commendable effort in terms of the data scale. However, the study has several limitations that significantly impact the confidence in the resulting national rainfall erosivity estimates:**

**(1) Snowfall Consideration: The study does not discuss how snowfall during precipitation events is treated. It is important to clarify whether snowfall contributes to rainfall erosivity, and if so, how it was incorporated into the model, as this could influence the overall erosivity estimates.**

Currently, no national monitoring system reports snowfall events in Mexico (Soto & Delgado-Granados, 2022). The National Center for Disaster Prevention developed a national Snow danger index at a municipality level, where only six municipalities from the northern States of Baja California, Chihuahua, and Durango are reported as high-risk. Data with greater detail is still lacking in Mexico.

To estimate the surface covered by snow, we could explore the climate classification developed by García (1998), a modified Köppen classification system (KCm) to better fit Mexico's climate conditions. According to the KCm, only 83 km2 can be classified as E climates in the format of ET (tundra: temperature of warmest month greater than 0 °C but less than 10 °C) and EF (snow/ice: temperature of warmest month 0 °C or below), both only produced by high altitudes.

The methodology section of the next manuscript will explain why we are not estimating rainfall erosivity by snow.

García, E. - Comisión Nacional para el Conocimiento y Uso de la Biodiversidad (CONABIO), (1998). 'Climas' (clasificación de Koppen, modificado por García). Escala 1:1000000. México.

Soto, V., & Delgado-Granados, H. (2023). Occurrence and characteristics of snowfall on the highest mountain of Mexico (Citlaltépetl volcano) through the ground's surface temperature. Atmósfera, 38, 35–54.

**(2) Transferability of Parameters: The study uses the parameters from Xie (2016), which were derived for China, to estimate rainfall erosivity in Mexico. Given the substantial differences in rainfall patterns and climatic conditions between these two countries, the direct application of these parameters raises concerns. A more thorough justification for the transferability of these parameters is needed, or alternative, region-specific parameters should be considered.**

We used the calibration parameters used in Xie et al. (2016) based on different reasons:

- Different review papers have shown that China extensively studies soil erosion and rainfall erosivity (Zhao et al., 2024) due to the fragile ecosystems that cause the most soil-erosion-affected area in the world: the Loess Plateau of China. This statement has caused an increasing interest in research on soil erosion in China, which has led to the development of extensive studies about erosion factors, such as rainfall erosivity.
- The World Meteorological Organization recommends studying climatic conditions using a time series for at least 30 years (considered a climatic normal). The calibration parameters from Xie et al. (2016) were estimated using rainfall time series between 31 and 40 years of registers.
- They found no geographic pattern in the beta variation within ten weather stations across China (latitudes between 25° and 50°).
- In the model used for our study (equation 1), the alpha coefficient varies as a sinusoidal function of the month, as proposed by Yu and Rosewell (1996).

However, the reviewer´s suggestion to use a region-specific parameter will be attended to. Following the Richardson et al. (1983) power law equation, a paper review made by Wang et al. (2024), they compiled the values for alpha and beta parameters from different climatic regions across the world (Based on Koppen-Geiger classification). We will calculate the R-factor again according to the parameters calibrated by each climatic region in Mexico. We will compare the erosivity estimations and complete our discussion.

W. Richardson, C., R. Foster, G., & A. Wright, D. (1983). Estimation of Erosion Index from Daily Rainfall Amount. Transactions of the ASAE, 26(1), 153–156. https://doi.org/https://doi.org/10.13031/2013.33893

Wang, L., Li, Y., Gan, Y., Zhao, L., Qin, W., & Ding, L. (2024). Rainfall erosivity index for monitoring global soil erosion. CATENA, 234, 107593. https://doi.org/https://doi.org/10.1016/j.catena.2023.107593

Xie, Y., Yin, S., Liu, B., Nearing, M. A., & Zhao, Y. (2016). Models for estimating daily rainfall erosivity in China. Journal of Hydrology, 535, 547–558. https://doi.org/https://doi.org/10.1016/j.jhydrol.2016.02.020

Yu, B., & Rosewell, C. (1996). A Robust Estimate of the R-Factor for the Universal Soil Loss Equation Author. American Society of Agricultural and Biological Engineers, 39(2), 559–561.

Zhao, Y., Zhu, D., Wu, Z., & Cao, Z. (2024). Extreme rainfall erosivity: Research advances and future perspectives. Science of The Total Environment, 917, 170425. https://doi.org/https://doi.org/10.1016/j.scitotenv.2024.170425

**(3) Validation Limitations: While the author attempts to validate the results using GloREDa data, the validation is based on a limited dataset of only 15 sites, covering a short time span, and restricted to the mountainous region of Michoacán. Given this narrow scope, the validation does not provide sufficient confidence in the accuracy of the national rainfall erosivity estimates. A broader validation across different regions and time periods would strengthen the findings.**

We understand that using a large dataset of rainfall time series with finer time resolutions (sub-hourly) will better validate our erosivity estimations. However, as in the Latin American and Caribbean region (LAC), Mexico faces the challenge of a weak climate monitoring network, where there are common issues with climate information such as low density or lack of in situ observations, scarce and intermittent time series at a fine scale (sub-daily), and a lack of FAIR (Findability, Availability, Interpretability and Reproducibility) principles of climate data (Cavazos et al., 2024). This last issue is principally due to the lack of open data policies, data quality control, and procedures for data collection that do not necessarily follow the standards of the World Meteorological Organization (WMO).

In this research, we tackled some challenges, such as the lack of quality control in rainfall time series and FAIR principles of climate data. Also, we present a potential use of this new database to address the most significant soil degradation problem: water erosion. We couldn't validate our erosivity estimations with a sub-hourly time series because the finest public, national, and available climatic data is at a daily resolution. For that reason, we presented a verification by comparing our estimations against three independent datasets as follows:

1. **GloREDa product**. The first verification was performed by using seven ecoregions as spatial support. We calculated the mean value of the erosivity for our dataset and the erosivity surfaces published by Panagos et al. (2023). In the case of the GloREDa surface, we calculated the mean value by using all pixels inside each ecoregion (Figure 8a in the manuscript).

2. **National Dataset (Erosivity-Cortés dataset)**. The second verification was performed using a national dataset developed by Cortés (1991). They used 42 rainfall time series with at least five years of registers at a time resolution of 1 minute. In this case, we plotted the relationship between rainfall erosivity and Annual rainfall for Erosivity-Cortés and Mexico-CN1(1968-1997) and Mexico-CN2 (1978-2007) datasets. We found a similar slope of 5.95, 5.47, and 5.41 for erosivity-Cortés, Mexico-CN1, and Mexico-CN2, respectively.

3. **Local Dataset (Erosivity-Michoacán)**. As a finer rainfall time series (15 minutes) was available for this region, we calculated the R(EI30) factor as defined in Wischmeier and Smith (1978) and the R factor by using the calibration coefficients (Xie et al., 2016) with the aggregated series (1 day). For these local conditions, we found that for every unit change in R with Xie's equation, the R(EI30) is 1.85 times larger, representing a large difference. However, we highlight that the large difference is for very local precipitation conditions and can not be generalized for all Mexican territory.

However, as suggested by Referee two, we could include a comparison between the erosivity values of the 15 Mexican weather stations in GloREDa and the nearest weather stations in our database (Mexico-CN3 1988-2017). Additionally, we could compare the erosivity values of Erosivity-Cortés dataset and the nearest weather station in our dataset (Mexico-CN1 and Mexico-CN2).

Cavazos, T., Bettolli, M. L., Campbell, D., Sánchez Rodríguez, R. A., Mycoo, M., Arias, P. A., Rivera, J., Reboita, M. S., Gulizia, C., Hidalgo, H. G., Alfaro, E. J., Stephenson, T. S., Sörensson, A. A., Cerezo-Mota, R., Castellanos, E., Ley, D., & Mahon, R. (2024). Challenges for climate change adaptation in Latin America and the Caribbean region. Frontiers in Climate, 6. https://doi.org/10.3389/fclim.2024.1392033

Cortés Torres, H. G. (1991). Caracterización de la erosividad de la lluvia en México utilizando métodos multivariados. Colegio de Posgraduados - Institución de Enseñanza e Investigación en Ciencias Agrícolas.

Panagos, P., Hengl, T., Wheeler, I., Marcinkowski, P., Rukeza, M. B., Yu, B., Yang, J. E., Miao, C., Chattopadhyay, N., Sadeghi, S. H., Levi, Y., Erpul, G., Birkel, C., Hoyos, N., Oliveira, P. T. S., Bonilla, C. A., Nel, W., Al Dashti, H., Bezak, N., … Borrelli, P. (2023). Global rainfall erosivity database (GloREDa) and monthly R-factor data at 1 km spatial resolution. Data in Brief, 50, 109482. https://doi.org/https://doi.org/10.1016/j.dib.2023.109482

Wischmeier, W. H., & Smith, D. D. (1978). Predicting Rainfall Erosion Losses: A Guide to Conservation Planning. Handbook 537. U.S. Department of Agriculture.

**Referee Comments 2**

**A well structured manuscript which address the important topic of rainfall erosivity in a high erosive country as Mexico. However, I will propose a major revision.**

**Authors did not discuss about the discrepancies (and uncertainties) when R-factor is calculated based on daily data. As Rainfall erosivity is much dependent on the intensity and not only on the duration, authors should discuss this shortcoming of their study. You are advised to read how the Rainfall Erosivity Database at European Scale (REDES) and the Global one (GloREDa) have been developed and why an approach of high temporal rainfall resolution (15 min, 30 min, 60 min) was chosen. In addition, the correct estimate of erosivity based on amount and intensity should be also reflected in the introduction (L 50-60).**

We agree with the reviewer. We understand that using finer time resolutions (sub-hourly) will better estimate erosivity values since the erosivity factor (EI30) is initially calculated by storm events (Renard et al., 1997; Wischmeier & Smith, 1978). However, as recognized by different authors, the scarcity of high temporal resolution (sub-hourly) climate data limits the estimation of the EI30 factor (Nearing et al., 2017). Various approaches to estimate the R-factor have been used to approximate the EI30 factor, as we did in our research. We will expand our discussion about the limitations of using a coarser resolution (daily) in erosivity estimation.

Nearing, M. A., Yin, S., Borrelli, P., & Polyakov, V. O. (2017). Rainfall erosivity: An historical review. CATENA, 157, 357–362. https://doi.org/https://doi.org/10.1016/j.catena.2017.06.004

Renard, K. G., Foster, G. R., Weesies, G. A., McCool, D. K., & Yoder, D. C. (1997). Predicting Soil Erosion by Water: A Guide to Conservation Planning With the Revised Universal Soil Loss Equation (RUSLE) (Agricultur). USDA United States Department of Agriculture.

Wischmeier, W. H., & Smith, D. D. (1978). Predicting Rainfall Erosion Losses: A Guide to Conservation Planning. Handbook 537. U.S. Department of Agriculture.

**In their comparison with GloREDa, authors sum the 12 monthly erosivity maps and compare their assessment with the summed map. It would be wiser to compare your results with the GloREDa dataset (map) that was produced in 2017. The objective of the 12 monthly erosive months maps was not to have a Global annual erosivity map which already exists since 2017. This should be carefully addressed both in the introduction, in L180-185 and in the comparison of results.**

We agree with the reviewer´s comment. We will compare it to the "Rainfall Erosivity in the World" performed by Panagos et al. (2017).

Panagos, P., Borrelli, P., Meusburger, K., Yu, B., Klik, A., Jae Lim, K., Yang, J. E., Ni, J., Miao, C., Chattopadhyay, N., Sadeghi, S. H., Hazbavi, Z., Zabihi, M., Larionov, G. A., Krasnov, S. F., Gorobets, A. V, Levi, Y., Erpul, G., Birkel, C., … Ballabio, C. (2017). Global rainfall erosivity assessment based

on high-temporal resolution rainfall records. Scientific Reports, 7(1), 4175. https://doi.org/10.1038/s41598-017-04282-8

**A proper verification/validation takes place against measured erosivity with high temporal rainfall records (less than 1h). You could make a verification by comparing your results as stations with closest (in distance) stations of GloREDa.**

It will be included. Additionally, we will consist of the same proposed comparison for the 42 weather stations in the Erosivity-Cortés dataset.

**In the discussion, it would be useful to mention the use of your dataset to identify trends in erosivity (what are the current trends in the last 50 years?) and how your dataset can be used to make projections of erosivity in 2050 and 2070 based on climate change scenarios?**

This discussion will be included in the manuscript.

**L73-75 are not necessary**

**Please refer to the density of the stations . One station per Km2?**

It will be included.

**L100-110 can be mode densed (concise) described .**

It will be corrected.

**In the abstract it should be mentioned that your measured input data are daily.**

It will be included.

**L5 "moments" is not the right word.**

 It will be corrected.

**Please use less acronyms in the abstract.**

It will be corrected.

**In Fig. 3 you can also add the locations of the 15 stations of GloreDa.**

It will be included.